# What contributes to good outcomes? The perspective of young people on short-term psychoanalytic psychotherapy for depressed adolescents

**Harriet Housby** [1¤], **Lisa Thackeray**[1,2], **Nick Midgley** [1,2]*

**1** Research Department of Clinical, Educational and Health Psychology, University College London, London [UCL], United Kingdom, **2** Child Attachment and Psychological Therapies Research Unit [ChAPTRe], Anna Freud Centre, London, United Kingdom

¤ Current address: Population, Policy and Practice Research and Teaching Department, UCL Great Ormond Street Institute of Child Health, London, United Kingdom
* nicholas.midgley@ucl.ac.uk

**Data Availability Statement:** The submitted paper concerns human research participant data which we will not make freely publicly available in an identifiable format because the original study

## Abstract

Depression is the fourth leading cause of adolescent illness and disability worldwide. A growing evidence base demonstrates that Short Term Psychoanalytic Psychotherapy [STPP] is an efficacious treatment for moderate to severe adolescent depression. However, with research in its infancy, key factors contributing to efficacy are unknown. Service users' lived experiences provide valuable insight in this area. This study aimed to elucidate what adolescents value in treatment by inductively exploring lived experiences of STPP. Five adolescents with the largest reduction in depressive symptoms scores between baseline and end of treatment, who had taken part in a large-scale randomized controlled trial, were sampled. In-depth interviews carried out soon after the end of therapy were analysed using Interpretative Phenomenological Analysis. Three superordinate themes were identified: "Therapy as a Transformational Process", "Explorative and Exposing: The Therapeutic Space" and "Being Heard and Working Together: The Therapeutic Relationship". Adolescents valued a process of collaborative exploration with the therapist which when it was achieved was felt to facilitate a deep-rooted transformation in self-perception. Additionally, they described how an adjustment was needed to the particular frame of a psychoanalytic therapy. However, not all participants with a good treatment outcome experienced therapy in this way, suggesting a potential gap between the quantitative assessment of outcomes, and the way young people experience and understand the change process. Clinical implications and directions for research are discussed.

## Introduction

Depression is the fourth leading cause of adolescent illness and disability worldwide [1]. Of adults who have experienced depressive disorders, more than half report first experiencing

information sheet, the terms under which young people consented to participate, did not include this data sharing, neither did our UK data protection legislation compliant data processing notice. We will consider requests from accredited researchers to share the relevant excerpts from the transcripts. Any such requests should be sent to DPO@annafreud.org quoting reference IMPACT Me PLOS ONE, in the first instance.

**Funding:** The authors received no specific funding for this work.

**Competing interests:** The authors have declared tha no competing interests exist.

depression before the age of fourteen [2]. For those who have a first episode of severe depression during their teenage years, there is a high risk of recurrence and a heightened risk of both intentional self-harm and suicidal ideation [3]. Although there is some evidence for the effectiveness of a range of psychological interventions [4], much remains uncertain about what treatments are most effective for which young people, and there is still a lack of understanding of the processes underpinning different psychological therapies.

Psychoanalytic theories of depression go back more than one hundred years [5], and therapies based on psychoanalytic principles now have a strong evidence base in the treatment of adults [6]. However, the evidence-base for psychoanalytic work with depressed children and adolescents has been slower to develop [7]. Starting in the 1990s, some evidence from naturalistic and retrospective evaluations began to emerge [8, 9]. Trowell et al. [10] published the first multi-centre randomized trial to assess a time-limited psychodynamic therapy for child and adolescent depression. In comparison to a systems integrative family therapy, the study found significant reductions in disorder rates for both groups [10], with good outcomes being maintained six months following the end of treatment. There was also an overall reduction in co-morbid conditions for young people in both arms, including improvements in family functioning, self-esteem and social adjustment [11, 12].

The promising findings of the Trowell et al. [10] study have been supported by a more recent clinical trial. As the largest and only fully-powered randomized clinical trial [RCT] assessing the medium-term clinical- and cost-effectiveness of short-term psychoanalytic psychotherapy [STPP] for depressed adolescents to date, the Improving Mood Through Psychoanalytic and Cognitive-Behavioral Therapy [IMPACT; 13, 14] study design strengthens the evidence base for this mode of treatment. IMPACT was a multi-site RCT comparing the efficacy of three psychological therapies for adolescent depression. Four hundred and sixty-five adolescents with moderate to severe depression were recruited from clinics across the UK and randomized to either STPP, Cognitive Behavior Therapy [CBT] or a Brief Psychosocial Intervention [BPI]. Results demonstrated that the therapies were equally efficacious at maintaining reduced depression scores one-year post-treatment and no significant differences were found in cost-effectiveness. As a result of these studies, the UK government included short-term psychoanalytic psychotherapy as one of a range of evidence-based treatments for child and adolescent depression in their clinical guidelines [15].

STPP assumes that our behavioural and emotional responses are meaningful and logical as, rooted in our internal worlds, they are based on our early experience of relationships [16]. STPP aims to explore these connections through a focus on the transference in the therapeutic relationship. Transference occurs in therapy when a client 'projects' past relational experiences or internal working models of relationships onto the therapist. Thus, the therapeutic relationship is seen as a window to past relationships and current expectations about self and other, and a safe context in which to explore and respond to unprocessed emotions [17, 18]. Through exploring these connections, the therapist can help the adolescent challenge patterns of relating to themselves and others and develop greater emotional insight and awareness. Busch, Rudden and Shapiro [19] state that 'good/successful outcomes' of STPP include effective management of depressive and aggressive feelings, a better sense of agency and a capacity to be thoughtful rather than "act out" emotions. Equipping adolescents with 'tools' such as insight and a capacity for reflection are thought to contribute to the prevention of recurrent symptoms later in life [20].

Manualised STPP consists of 28 weekly sessions comprised of three treatment stages [17]. The initial stage focuses on establishing a framework for therapy [confidentiality, and time-limited nature of the work], forming a therapeutic alliance and exploring the adolescent's reflective capacity. The purpose of this stage is to help the adolescent realise that their

symptoms are connected to thoughts and feelings and hold meaning. The middle stage of therapy facilitates a more in-depth exploration of symptom meaning, which fosters a greater capacity for the adolescent to confront problematic relationships to the self and others. Attending to the transference relationship, the therapist selects and, in a way that makes sense to the adolescent, explores aspects of the adolescent's verbal and non-verbal unconscious communication. The final stage focuses on progress, achievements in therapy, and separation from the process.

## Exploring the adolescent's perspective

While the evidence base for the effectiveness of STPP grows, less is known about how it leads to change [14, 21]. Although there have been some attempts at 'dismantling' studies to assess the impact of specific aspects of therapy [such as the use of transference interpretations, e.g. [22], service users' lived experience provides invaluable insight not only into the effectiveness of therapeutic treatments but also the way in which young people give meaning to their experiences and understand what contributed to change [21, 23, 24].

A number of qualitative studies have examined how young people understand what contributes to therapeutic change, including in BPI and CBT [25, 26]. However, Løvgren et al. [27] were the first to qualitatively explore depressed adolescents' experience of change through STPP. Eight females participating in the First Experimental Study of Transference Work–In Teenagers [FEST-IT] were interviewed following 28 weeks of STPP about their experience of therapy. These semi-structured interviews were analysed using systematic text condensation [STC]. The adolescents experienced improvements such as increased insight through an exploration of the self, relationship with the therapist, integrating therapy into everyday life and developing adequate internal working models of relationships over time. Similarly, Marotti, Thackeray, and Midgley [28], focusing specifically on the experience of teenage boys in STPP, identified the important role of the therapeutic relationship and the ability to gain self-understanding as essential elements of the change process. However further studies are required to provide a more nuanced account of adolescents' experience of improvement through STPP.

This study therefore aims to extend our understanding of what contributes to therapeutic change in the psychoanalytic treatment of depressed adolescents, by exploring the experience of adolescents who had good outcomes in the STPP arm of the IMPACT trial. By focusing only on those who had good outcomes, it aims to help build a model of therapeutic change in short-term psychoanalytic treatment, from the perspective of those who had directly participated in the therapy.

## Methods

### Study design

This study was carried out using Interpretative Phenomenological Analysis [IPA] [29]. The theoretical underpinning of IPA allows the researcher to explore how participants make sense of and ascribe meaning to their experiences while recognising that their own personal, professional and cultural experiences affect the analytic process [30]. IPA seeks an in-depth understanding of both shared and idiographic aspects of a phenomenon in a specific context, thus giving a detailed insight into participants' converging and diverging experiences.

Previous studies linked to the IMPACT trial have explored adolescents' experience of CBT [26] and BPI [25]. These studies both made use of IPA to examine the experience of young people with good outcomes in the clinical trial. The current study chose to use a similar approach to explore the experience of young people in the STPP arm, to improve

understanding of unique and shared treatment factors that may guide refinement of models for adolescents with moderate to severe depression.

## Setting

This study draws on data from IMPACT My Experience, [IMPACT-ME] [31], a qualitative sub-study nested in the IMPACT RCT [14], which aimed to examine the experience of parents, young people and therapists who had been involved in the North London region of the clinical trial. The IMPACT-ME study included semi-structured interviews conducted at three time points: baseline [T1], end of therapy [T2 ~ 36 weeks], and, 1-year post-therapy [T3 ~ 86 weeks] [for full details, see [31]].

The current study draws on interviews with adolescents from the STPP arm of the 'IMPACT-ME' project [31]. In the context of the IMPACT trial, Short-Term Psychoanalytic Psychotherapy [STPP] was a manualised treatment, offering up to 28 sessions over a 30-week period [17]. Like other psychoanalytic and psychodynamic treatments, the STPP approach assumes that our behavioral and emotional responses are meaningful and logical as, rooted in our internal worlds, they reflect our early experience of relationships [16]. STPP aims to explore these connections through working with the transference and countertransference in the therapeutic relationship. The therapeutic relationship is seen as a window to explore internal working models of relationships and a safe context in which to explore and respond to unprocessed emotions [17, 18]. Through exploring these connections, the STPP model postulates that the therapist can help the adolescent challenge patterns of relating to themselves and others and develop greater emotional insight and awareness. Equipping adolescents with 'tools' such as insight are thought to contribute to the prevention of recurrent symptoms later in life [20].

## Sampling and participants

44 adolescents randomized to the STPP arm of the IMPACT trial were interviewed as part of the IMPACT-ME study. Of these, fourteen were excluded because of missing data for the Mood and Feeling Questionnaire [MFQ; 32]. Because of the in-depth nature of the data analytic process, it is common for IPA studies to have 4–6 participants [29], and for the sampling approach to focus on homogeneity in relation to the research topic. Typically, five participants is recommended as a sample size, because it allows for a balance between in-depth analysis with the possibility of exploring a range of experience of a particular phenomenon [29]. Thus, of the remaining 30, those five with the greatest point score reduction on the MFQ between T1 and T2 were sampled. This produced a sample of 4 females and 1 male aged 14.4–17.7 years at baseline [M = 16.0, SD = 1.53]. The sample were fairly representative of all STPP cases in the IMPACT study, which was 74% female, with a median age at baseline of 15 years. The mean MFQ score at baseline for young people in the STPP arm of the IMPACT study was 45.4, compared to 43 for the participants in this study. Their pseudonymized details can be found in Table 1.

## Data collection

All participants were interviewed soon after the end of treatment [T2] by a trained research psychologist, using the semi-structured "Experience of Therapy" interview schedule [33]. The interview schedule consists of open, non-directive questions aiming to elicit in-depth exploration of the adolescent's story of depression, their experience of therapy, understanding of what has changed between ending therapy and time of the interview, and experience of their

**Table 1. Participant demographics: Age, number of sessions offered & attended, Mood and Feelings Questionnaire [MFQ] scores at T1 and T2.**

| Participant | Gender | Age [T1] | Age [T2] | Sessions offered | Sessions attended | MFQ score at baseline [T1] | MFQ score at end of therapy [T2] |
|---|---|---|---|---|---|---|---|
| Mitch | Female | 17.7 | 18.8 | 28 | 24 | 54 | 12 |
| Joey | Male | 15.3 | 16.5 | 28 | 14 | 50 | 14 |
| Talitha | Female | 17.7 | 18.5 | 28 | 20 | 32 | 7 |
| Selah | Female | 14.4 | 15.3 | 20 | 9* | 37 | 14 |
| Anaya | Female | 15.3 | 16.3 | 28 | 26 | 42 | 25 |

*Selah is coded on the IMPACT database as therapy 'drop out'; All others are coded as 'completed'.

participation in the research study [31]. Interviews were carried out at a location chosen by the young person [usually their home or the clinic] and were audio-recorded and transcribed.

## Data analysis

Data analysis adhered to IPA protocol; Table 2 describes the iterative analytic process [34]. A process of free coding and phenomenological coding set the foundations for initial theme identification and clustering. Superordinate themes and subthemes were identified before conducting final cross case analysis to form the final sub-and superordinate themes. Themes in this study were selected due to their relevance to the research question and were refined during the writing process [29].

Initial data analysis was carried out by the first author [HH]. To increase the credibility and dependability of the analysis, the input of peers and an experienced supervisor [LT] were sought to test analytic decisions and develop interpretations. Further, large data extracts from the participants' interviews were included in the write up to support analytic claims. The feedback also facilitated a reflexive awareness of the researcher's position towards the data and the experiential lens through which it was viewed. This enabled the researcher to engage with potential biases in interpretation.

A reflexive diary was used to document the process of analysis. Thought processes, essential extracts and themes identified at each stage [30] in addition to reflections on beliefs, perceptions and personal experiences which may have impacted the analytic process were noted.

**Table 2. Data analysis steps.**

| | |
|---|---|
| 1 | **Free coding:** Transcripts read alongside corresponding audio recording to facilitate data emersion. Annotations of phrases, psychological concepts, and emotional reactions initially salient to the researcher were made to identify and minimise the impact of preconceptions on the analytic process. |
| 2 | **Phenomenological coding:** Using clean transcripts, descriptive summaries of participants' experience were noted in the left-hand margin. In the right-hand margin initial interpretations of these experiences were made using succinct phrases to capture meaning. This enabled a close engagement with the data and interpretations grounded in the participants' experience. |
| 3 | **Initial theme identification and clustering:** Summaries most significant in understanding the participants' experience of STPP were copied onto post-it notes to visualise emerging connections. These were iteratively clustered to form subordinate themes incorporating the researcher's interpretation of the data and the interviewee's words. |
| 4 | **Superordinate and subtheme identification**: The subordinate themes were clustered, and superordinate themes identified, labelled and tabulated with supporting extracts from the participants' interview. |
| 5 | **Cross case analysis:** The thematic tables from each participant were reviewed. Subordinate themes for all the participants were clustered to form the final sub-and superordinate themes which best captured shared convergence and divergence across the sample. |

### Ethical considerations

The study was approved by the Cambridgeshire 2 Research Ethics Committee [reference 09/H0308/137] and local NHS provider trusts. All patients and their parents gave written informed consent [31]. This study complies with ethical requirements. Pseudonyms ensure the participants' anonymization, and identifiable information within the transcripts, such as names of towns or clinics, were replaced with codes: [CLINIC] [TOWN].

## Results

Three superordinate themes were identified during this analysis [Fig 1]. These themes cover six subordinate themes that portray the adolescents' convergent and divergent experiences of STPP. All five adolescents contribute to the superordinate themes, a minimum of three contribute to each subtheme. Extracts from the adolescents' interview data illustrate each of the themes to ensure transparency in the analytic interpretations [34].

### Theme 1: Therapy as a transformational process

The first theme encapsulates the adolescents' shared experience of engaging and then beginning a process of transformation throughout therapy.

**"just sort of grew on me more": Adjusting to the therapeutic model.**   Joey, Anaya and Talitha reported a shared experience of slowly finding security in the therapeutic process. All three experienced feelings of shyness and inadequacy at first, trying to find "the right answer" [Joey], before feeling more able to express themselves. Talitha describes putting aside her expectations of the therapy as interactive, gradually adapting to the reflective space STPP provided:

**Fig 1. Description of superordinate and subordinate themes encapsulating adolescents' experience of STPP.**

"I came to therapy thinking that. . .the focus would be a bit more, sort of, I dunno, inter-active. . . giving you advice, but I knew–but then I realised that therapy isn't really a place to be given advice, it's just a place to like–you to let your feelings out and then go—like–take that and then go out with it knowing that you let your feelings out and you can move on" **Talitha**

Joey also described adjusting to STPP's "formal" structure and "awkward silences" which contrasted with his previous experiences of child and adolescent mental health services [CAMHS]. However, he experienced a smoother transition to STPP than Talitha. Joey received drug, school, crisis, and private counselling alongside STPP. He described each therapy as hav-ing its own function, working in a "perfect mixture" that allowed him to process difficulties. Indeed, throughout his transcript, Joey appeared more open-minded and engaged with the explorative nature of the model than other adolescents.

Mitch and Selah had divergent experiences of adjustment to a psychoanalytic therapy com-pared to the other three participants. Mitch did not experience therapy as increasingly com-fortable but rather endured it. She returned to sessions intending to 'find the source' of her symptoms. Although aversive, she adjusted to make use of STPP in her own way:

it was really awkward coz like I-I just we'd spent the whole time just sat there saying noth-ing, so it was really kind of just a-a goodbye kind of thing. . . **Mitch**

By contrast, although Selah had a 'good' outcome in terms of changes in her MFQ scores, she terminated therapy after nine sessions, reporting feeling "upset and depressed" after each meeting to the point where she avoided attending sessions. Ultimately, her experience was marred by the suggestion of social service involvement:

"when he started saying about bringing social services in, I just completely stopped. That was just like the end of the line for me" **Selah**

Selah and Mitch's experiences make clear that not all adolescents–even those with appar-ently 'good' outcomes –necessarily adjusted to the particular expectations of this therapeutic model. Their more negative experiences support this idea that an adjustment to the particular frame of a psychoanalytic therapy–including a capacity to 'let feelings go' and being open to process difficulties without necessarily being given direct advice–is a core element of the thera-peutic process for depressed teens in short-term psychoanalytic therapy.

**"It transforms you": Gaining a new perspective.** All adolescents experienced STPP as facilitating a change in perspective that improved their lives. Anaya realised that her contribu-tions, and by extension herself, were "significant" as 'boring' topics of friendships and roman-tic experiences became "interesting or useful" once they were explored in therapy. Similarly, Joey reported that therapy "woke [him] up" to how he reacted in situations. He described a process of reframing his thoughts, as links between events "slowly started to unravel". Whereas before he was impulsive, he described how engaging with STPP helped him understand and "always read [him]self" better:

"That's usually my big issue. I don't know why I do things. . . I'm very impulsive. That's what my tutor calls me anyway. An impulsive person. . .so I just do things and not think about it. . . then I sort of. . .I left counselling most of the time I would sit there goin' 'do you know what, that makes sense why I did that'" **Joey**

Similarly, Talitha described experiencing 'transformation' through therapy. She explained that STPP was "a very helpful process" that changed her thought patterns, and described learning to express herself without fear, and to be more open with her family. Overall, she described therapy as allowing her to "come out stronger" and experience "being freed from a trap":

"I am very positive nowadays before I was quite like negative. So yeah, I guess, like now, after the bullying, joining IMPACT centre, I think this changed, not just IMPACT but also the counselling has like changed, sort of the way in which I am" **Talitha**

Mitch and Selah also experienced a similar transformation in perspective. Mitch explained that she realised that although 'bad' events such as rejection happen, it was 'good' that she could express loving feelings. Similarly, Selah felt "stronger" after therapy. However, their transformation experience was different from the other adolescents. They appeared to engage less in therapy and felt that they had found their own solutions to their depression. Mitch, spent most sessions in silence and "snapped out" of her depression:

"I don't really feel like that obviously like I was depressed, but it's kind of thinking why- what do I need to be depressed for [. . .], and it kind of snapped me out of it really. . ." **Mitch**

Mitch appeared to take control of her therapy acting with autonomy, rather than working with the therapist to 'resolve' her depressive symptoms. Similarly, Selah exercised autonomy by terminating therapy. Experiencing the therapy as intrusive, she sought a friend's support.

## Theme 2: Explorative or Exposing: The therapeutic space

This second theme describes the adolescents' experiences of the sessions themselves. Four adolescents experienced therapy as an explorative space allowing them to express and make sense of their depressive symptoms. All adolescents described experiencing some level of exposure, which left two participants feeling inadequate or unsafe.

**"slowly starting to unravel": An explorative space.**   Joey emphasised the value of the exploration in STPP. He viewed it as an "essential level" of therapy, working synergistically with more action-focused interventions, such as drug counselling. He described how the STPP provided him with an insight into his psyche so that other therapies felt less threatening:

"I could sit there in IMPACT and 'oh right so this is possibly why [. . .] this was happening'. . ..then other counselling helped coz I could learn how to fix it. . . That's sort of what I think the three different counsellors all put together. . .with the sort of perfect mixture, but definitely without IMPACT it would've been a sort of stab in the dark situation" **Joey**

Joey described therapy as "waking up" "understanding his brain" and his "subconscious". He gave this example:

"I got a G in my favourite subject, and [the therapist] were like 'right. . .and the next day you self-harmed'. . .I was like yea. . .and to me that doesn't make. . . that's just like G self-harm. . .but then it's like 'how did you feel about the G', and I thought I didn't feel anything but then sort of slowly starting to unravel and I was like 'you know what that kind of makes sense'" **Joey**

Talitha described therapy as a space to "get it all out of your system" like a "toilet". As therapy progressed, the therapist challenged her to bring her emotional, 'toilet self' and 'thinking self' to sessions. This allowed her to bring her whole self and explore her feelings by "putting them on a plate" to "organise" them. Building on her own metaphor, it was as if she was reframing her feelings from 'waste' to 'food', both digestible and nutritious.

Despite her more negative experience of therapy, Mitch described her experience of "awkward silences" as "really helpful". Even though she did not like the silences, she explained how they allowed her to consider how much she wanted to engage with her difficulties, thus helping in an unexpected way. It appears they enabled her to find a thinking space, where she made her own decisions about engaging more fully with the difficulties she was experiencing.

**"the more I thought**. . .**the more I panicked": An exposing space.** Although most participants described the way STPP encouraged them to explore, the non-directive element could also be experienced more negatively. Joey, Anaya and Talitha all spoke of initial feelings of exposure in the form of inadequacy:

> "trying to find something that I could talk about and it just the kind of the more I thought about it, the more I kind of I dunno panicked well not panicked but kind of worked myself up [. . .] it just wasn't very helpful" **Anaya**

Talitha experienced the silence as feeling unfamiliar and difficult to tolerate, appearing compelled to fill the space:

> "I would sort of repeat myself. Yeah, I guess, like, I would sort of say the same things quite a few times and just like fill up the space [. . .] it felt a bit awkward because like [. . .] I was expecting her to say something. Well, I was expecting her–but then I knew I couldn't blame her for that because she didn't, she didn't really know me. . .." **Talitha**

Mitch felt that she "wasted this poor woman's time" by 'just' sitting in extended periods of silence. She felt exposed and abandoned by the therapist's open-ended questions, stating she would not recommend therapy to other young people unless they "know what the problem is before [they] go". However, Selah voiced the most extreme feelings of fear:

> "I was like no, never ever turning up to CAMHS ever again. Every time I walk past it I just get a shiver down my spine, in the politest way to say it, I wanna slap him for putting me in such a bad way for a while. . ." **Selah**

Describing it with a sense of anger, Selah described how her initial trust in the therapist felt betrayed once the therapist spoke about contacting social services, because of safeguarding concerns. Selah spoke about feeling very betrayed by this, having trusted the therapist enough to reveal her experiences. This feeling of exposure and betrayal led her to become very angry, and ultimately to end the therapy prematurely. Her experience suggests that when the sense of exposure outweighs a feeling of being supported, or decisions about safeguarding with adolescents are not felt to be made collaboratively, then therapy may break down.

## Theme 3: Being heard and working together: The therapeutic relationship

All the adolescents spoke about their experience with the therapist, and either implicitly or explicitly made clear how important they felt the therapeutic relationship was to any process of change.

**"Being heard": Building a trusting relationship.**    The extent to which each of the participants in this study felt heard and involved in the therapy appeared to influence their overall experience of STPP. Indeed, Joey spoke explicitly about the importance of "rapport" in aiding the therapeutic work:

> "but then sort of I started really getting along with. . .the person I had. . .and it was just a lot easier to talk. . .and to be honest it did actually sort of help so I'd walk away and I'd be like that made so much more sense. . ." **Joey**

Joey, Talitha and Anaya all experienced "being heard" by their therapists. Anaya described how the therapist used her own words to challenge her negative beliefs, and this allowed her to experience herself as significant:

> "they were saying like. . . 'why-why do you feel that it has to be something like really important?', 'why can't you I dunno talk about like something that you I don't know that you think is slightly insignificant?' like coz I think I used the word insignificant. . . I realised that. . . what starts as something boring. . . can actually be quite. . . interesting or useful" **Anaya**

Joey reported that being heard in a genuine way by the therapist made him feel understood. Similarly, Talitha described her therapist as 'very calm' and "a private friend" who she was able to "show the real me at my worst". She experienced the therapist as accepting, hearing and processing her words:

> "'coz I didn't know what to talk about, like, I would say something and then she would be like "Maybe that could be because of this or that" or something like that, so yeah. . .she goes like "work it out" and sort of, yeh. . .. it made me feel like she understands. . .like she is really taking in what I am saying" **Talitha**

These experiences contrasted with Mitch's and Selah's, who both felt judged by their therapists. Mitch reported 'awkwardly' repeating herself, perhaps reinforcing her feelings that she was wasting the therapist's time and suggested other adolescents seek out a friend as they "don't judge". Selah relied on her best friend during therapy for emotional support, experiencing her therapist as intrusive and repetitive:

> "I wouldn't like the same questions over and over again, I wouldn't like to be reminded of what I said before because when I get out, I tend not to think about it like it's just a blank thing but with him, he kind of reminded me 24/7 to the point where I just got fed up with it and I stopped so I would never turn to therapy again" **Selah**

Selah's experience suggests that for some young people being made to think about certain things, or reminded of them, was experienced as unhelpful. Whereas when therapists provide a space for the young person to feel understood they could discover meaning in experiences which has previously appeared to be meaningless.

**"we worked a way. . .to catch it": Working together.**    The adolescents in this study valued their therapists taking a collaborative approach. Joey spoke extensively of exploring issues together with the therapist and appeared to have internalised the therapist, feeling that he had greater insight into his mind because of therapy and could now "read himself":

"we had like a whole session talking about why it happened that was like what did I actually do, and we couldn't even—the woman who was like doing the counselling weren't really sure [. . .] we sort of sat there for a whole session [. . .] figured out, we worked a way about how not to overcome it but catch it in the bug bite sort of. . ." **Joey**

Conversely, Anaya and Selah did not experience therapy as a collaborative process. Anaya reported struggling to keep up with some of what the therapist said–'like we'd be talking about something quite general . . . and then if felt like he'd kind of gone deeper like three or four steps in thinking'. Despite eventually seeing the connections, she appeared to experience the therapist as taking on an expert role, in which all she could do was try and catch up:

"I hadn't kind of gone there with him [laughs] [. . .] We'd be here, and then he'd be there and I. . . I'd suddenly be there too, and I didn't understand how we got there [laughs]. . ." **Anaya**

This feeling that the therapist knew more than her could explain why Anaya was worried about "coping without therapy" as unlike Joey, who described how he learnt to 'read' himself, she felt more reliant on the therapist's guidance. Selah also experienced a power imbalance. However, unlike Anaya, she experienced the therapist as intrusive and deceitful, especially when the possibility of social service involvement was raised:

"I feel sorry for every child that has to go there, coz they're basically just backstabbers [. . .] like some counsellors yeah they might say right if anything gets out of hand we do have to say duh..duh..duh, to other people but if you don't say it to me then why should I trust you [. . .] I hate him" **Selah**

Selah's experience suggests that when there was not a sense of collaboration, then trust could easily be lost and mar the whole therapeutic process. This absence of a sense of collaboration in Selah's case suggests how important such a process was for a more successful therapeutic process to be achieved.

## Discussion

This study aims to extend our understanding of what contributes to therapeutic change in the psychoanalytic treatment of depressed adolescents, by examining the experience of adolescents who had good outcomes in the STPP arm of the IMPACT trial. IPA of semi-structured interviews identified three interrelated themes: 'Therapy as a Transformational Process', 'Explorative or Exposing: the Therapeutic Space' and 'Being Heard and Working Together: the Therapeutic Relationship'. These themes are discussed in turn, focusing on findings that are most informative for clinical practice and research.

The first theme emphasises the adolescents' experience of adjustment to STPP. Adolescents experienced an initial period characterised by feelings of inadequacy and altering expectations of therapy. Establishing a non-judgmental therapeutic setting appeared important in the initial stages of STPP [17], along with an adaptation to the particular kind of therapeutic approach being offered–one focused more on open exploration rather than direct guidance. Not all adolescents in this study experienced such an adjustment to STPP, and it appeared to be harder for those with no previous experience of therapy. Indeed, adolescents in this study who experienced a smoother adjustment to STPP had multiple previous experiences of therapy. This emphasises the importance of considering how past experiences of therapy may affect adolescents' capacity to engage with STPP [35].

These findings are echoed in previous work by Bury, Raval, and Lyon [35] who found that adolescents experience a period of "learning the ropes of therapy" when engaging in psychoanalytic treatment, setting aside their expectations of a pragmatic relationship to engage in a more open-ended and exploratory type of therapy. The desire for a pragmatic relationship could reflect the developmental conflict that defines adolescence—between agency and interdependence and an increased reliance on peers for emotional support [31, 36]. Through acknowledging this developmental conflict, the therapist may be able to create space for the adolescent to productively express their agency [37, 38]. Indeed, inviting the adolescent to contribute their understanding and expectations of therapy is vital in preventing ruptures that can be detrimental to the work [39].

Where participants described the impact of STPP in positive terms, adolescents emphasised transformation in their perspective–coming to see things in new and helpful ways. As in Løvgren et al. [27], some gained greater self-awareness through 'making-sense' of past events with the therapist. This is concordant with one aim of psychodynamic therapy, to challenge a client's internal working model of relationships and equip them with insight as a tool for long term improvement, the 'sleeper effect' [20]. Others in this study did not attribute their transformation to therapeutic activity but independent revelation.

The second theme captures the adolescents' experiences of STPP as explorative or exposing. In line with prior research, adolescents valued an 'explorative space' to make sense of past events and emotions [27]. STPP aims to facilitate this through a focus on the transference relationship [17]. In a 'container-contained' dynamic, the presence of the therapist allows the adolescent to safely explore conscious and unconscious thoughts and feelings [40]. The experience of some of the participants in this study appeared to support this model. One adolescent spoke of putting feelings on a plate, supporting Bion's [40] theory that the therapist receives, digests and returns emotion in a palatable format to facilitate insight. Considering that adolescents sometimes experience depression as a 'cutting off from the world' [41], this insight facilitated through exploring emotion was valued by some as an "essential level" of therapy.

Yet alongside this opportunity for more open-ended exploration, STPP could also be experienced in a more persecutory way. All adolescents in this study described feelings of exposure, particularly when faced by long silences [17]. Acheson et al. [42] have explored the role of silence in STPP with depressed adolescents and suggest that such silence may often be 'obstructive' to the therapeutic process. Whilst some participants in this study found it a helpful tool for reflection and expression of agency [38, 43], others experienced intense performance pressure, misunderstanding and abandonment [44]. In psychodynamic therapy there is some expectation of anxiety and a frustration of wishes; too little anxiety prevents meaningful insight; too much is overwhelming and unproductive [45]. For some adolescents unmanageable anxiety was evoked, which appeared to lead to a whole-scale rejection of the need for therapy. Finding the silence and focus on risk as a threat to their security their experience was synonymous with 'dissatisfied' psychoanalytic patients [46]. They disengaged by psychically cutting off from their symptoms or terminating therapy. This emphasises the importance of establishing a framework for therapy with depressed adolescents in the initial stages, in which some anxiety is accepted, but not to a degree to which it becomes overwhelming [17].

The final theme relates to the importance of the therapeutic relationship. Consistent with other successful psychoanalytic cases [47], all adolescents in this study emphasised the value of the therapeutic relationship in their experience of STPP. Indeed, one adolescent explicitly attributed insight to the good relationship they had with their therapist, supporting research that suggests a good therapeutic relationship fosters greater self-awareness [48].

Attentive listening, acceptance, curiosity and empathy were valued experiences that fostered such a relationship. For one adolescent, the therapist's use of the young person's own

language to challenge their beliefs facilitated a transformation in perspective [49]. However, where previous studies have suggested that all successful STPP cases with young adults experienced the therapeutic relationship as a secure base for exploration and change [47] some adolescents in this study experienced the relationship negatively, even if they could be described as 'good outcome' cases. Those who did not experience being heard found the therapist repetitive and intrusive leading to feelings of invalidation and frustration. Instead, they found these therapeutic qualities in peers and people outside therapy. It may be that in these cases the good outcome was not attributed by the adolescents to the therapy per se, and that this was because they did not find the therapeutic relationship a supportive one. This emphasises the importance of listening in building a trusting and productive therapeutic relationship [50, 51].

Collaboration was also highly valued by the participants in this study. 'Thinking together', is thought to create a space to simply 'be' that facilitates reflection [52–55]. It reduces power imbalances, demonstrated to negatively affect therapy [35, 56]. Acknowledging that both the therapist and adolescents are experts—the former in clinical knowledge and competence, the latter in their own experience, empowers the adolescent and helps to facilitate shared exploration [57, 58]. Those participants in this study who did not experience collaboration felt the therapist assumed an expert role. This promoted an overdependence on the therapist that conflicted with adolescent development and long-term therapeutic gain [59, 60]. A therapist who persists with their agenda may leave adolescents feeling unable to express themselves, leading to miscommunication [46]. One adolescent in this study experienced extreme feelings of betrayal when social services were mentioned. It is paramount that therapists set boundaries for the work, particularly addressing issues of confidentiality [17], as inconsistent expectations can lead to therapeutic ruptures [39]. Where such therapeutic ruptures do occur, it is important for those to be repaired [61]. Previous research using data from the IMPACT study has demonstrated that successful repair of ruptures early in therapy was one of the things that distinguished therapy completers from therapy dropouts [39].

## Clinical implications

Overall, it appears the adolescents with depression who are engaging in STPP most value an experience of collaborative exploration. When this was experienced, it facilitated a deep-rooted transformation in self-perception demonstrated in significant improvements in depressive symptoms. However, the disparity between quantitative outcomes and lived experience of therapy for some adolescents has interesting clinical and research implications.

From a clinical perspective, it emphasises the importance of collaboration. Where adolescents' voices have previously been limited [49], collaboration within therapy and on a service-wide level is now policy [62]. Involving the adolescent throughout therapy improves their experience by reducing power imbalance and promoting agency. Further, by creating a space for adolescents to express their expectations, preferences and understanding of STPP from an early stage, misunderstandings that commonly lead to dropout can be addressed [39]. In particular, where adolescents felt that they did not understand what was happening in therapy, or where long silences left them feeling uncertain and alone, this was mostly experienced in a negative way. Helping adolescents to tolerate an inevitable degree of uncertainly and anxiety, without leaving them with unmanageable levels, seems an important clinical implication of what these young people said about their experiences of STPP. The results of this study could be used to amend/enrich the STPPP protocol, especially with regards to the way the treatment is framed so that the adolescents can better understand what to expect. There is a growing body of literatures highlighting the importance of transparency and shared decision making in the therapeutic process [63]. If service users were better informed about what to expect in STPP as

part of the STPP protocol, could that mitigate the potentially hindering effect of anxiety and ease the adolescent's process of adjustment to the particular treatment mode.

From a research perspective, the finding that some young people with 'good' outcomes described quite negative experiences of therapy poses interesting questions about how the research community conceptualise and measure therapeutic success. Although it is possible that these participants did benefit from STPP, despite their negative experiences, a more parsimonious explanation would be that the improvements on their MFQ scores were related to factors outside therapy. This emphasises the importance of clinical trials not only assessing outcomes, but also building into the design of studies mechanisms to investigate the degree to which those outcomes can indeed be attributed to the therapy. There may also be a need for trials to capture a broader range of outcomes, beyond the usual focus on symptom change [64]. As this study demonstrates, attending to the experiences of service users adds an important element to our understanding of therapeutic outcomes.

## Strengths and limitations

The major strength of this study lies in the in-depth focus on the experience of those engaging in psychoanalytic therapy, given the lack of studies which have examined such an experience to date. IPA's idiographic approach means that participants less articulate about their experience of therapy are equally represented in the data, affording rich representation of convergence and divergence in theirs experience that is not possible using other methodology [27].

However, caution should be exercised when considering how the results transfer to other adolescents' experiences, both within the IMPACT study and in other settings. The fact that these therapies were offered as part of a clinical trial also limits the transferability of the findings. Firstly, adolescents were randomly assigned to this treatment modality whereas in routine clinical practice a shared decision would be made taking into account treatment goals and preferences [58, 65]. Secondly, adolescents, when interviewed about their experiences, often referred to IMPACT and STPP synonymously, without fully distinguishing between the research study and the therapy component. This implies a potential extraneous therapeutic effect of the research interviews on depression [66].

## Conclusion

In summary, adolescents with moderate to severe depression who engaged in psychoanalytic therapy as part of a clinical trial valued an experience of collaborative exploration with the therapist, which when it was achieved was felt to facilitate a deep-rooted transformation in self-perception. This involved a process of adjustment to the particular framework and processes of psychoanalytic therapy and could lead to a sense of expanded self-understanding. For those who did not experience this sense of collaborative exploration, however, aspects of the therapeutic relationship in STPP were experienced as aversive, meaning that change was attributed to factors outside therapy, or else that STPP 'helped but not in the way it should'. For those who did not experience this sense of collaborative exploration with the therapist, STPP was a place where they felt invalidated and forced to depend on themselves or seek others for support.

The adolescents' diverging experience emphasises that quantitatively good outcomes do not necessarily mean that the therapy itself should be understood as the cause of such outcomes. Thus, to improve our understanding of what adolescents value in therapy and the importance they attribute to it in their recovery, we must ensure that the perspective of young people is incorporated into future studies evaluating the effectiveness of a range of psychological therapies.

## Author Contributions

**Conceptualization:** Harriet Housby, Nick Midgley.

**Formal analysis:** Harriet Housby.

**Methodology:** Nick Midgley.

**Supervision:** Lisa Thackeray, Nick Midgley.

**Writing – original draft:** Harriet Housby.

**Writing – review & editing:** Harriet Housby, Lisa Thackeray, Nick Midgley.

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
