## [Decision Letter · Decision Letter 0]

9 Jun 2021

PONE-D-21-07634

What contributes to good outcomes? The perspective of young people on short-term psychoanalytic psychotherapy for depressed adolescents

PLOS ONE

Dear Dr. Midgley,

Thank you for submitting your manuscript to PLOS ONE. After careful consideration, we feel that it has merit but does not fully meet PLOS ONE’s publication criteria as it currently stands. Therefore, we invite you to submit a revised version of the manuscript that addresses the points raised during the review process.

Both reviewers were very fond of your work and contribution to enhance user utilization by understanding qualitative features of mechanisms of change and provided valuable and detailed feedback to further improve your manuscript.

We look forward to receiving your revised manuscript.

Kind regards,

Svenja Taubner

Academic Editor

PLOS ONE

Journal Requirements:

Reviewers' comments:

Reviewer's Responses to Questions

**Comments to the Author**

1. Is the manuscript technically sound, and do the data support the conclusions?

Reviewer #1: Yes

Reviewer #2: Yes

2. Has the statistical analysis been performed appropriately and rigorously? 

Reviewer #1: N/A

Reviewer #2: N/A

3. Have the authors made all data underlying the findings in their manuscript fully available?

Reviewer #1: No

Reviewer #2: No

4. Is the manuscript presented in an intelligible fashion and written in standard English?

Reviewer #1: Yes

Reviewer #2: Yes

5. Review Comments to the Author

Reviewer #1: The current study uses Interpretative Phenomenological Analysis (IPA) to qualitatively analyse post-treatment interviews with 5 adolescents to investigate their experience of short-term psychoanalytic therapy (STPP) for depression. The analysis gives valuable insight into which aspects are experienced as helpful and which aspects might be hindering a successful treatment for adolescents. With the authors’ thorough discussion, I find this a very important study shedding light on possible important aspects of successful therapeutic processes.

However, I would recommend several changes and elaborations:

Introduction:

- As the experience of the treatment is the focus of this study, I would find it helpful to elaborate more on the treatment (e.g. therapist stance, session structure)

Methods

Sampling:

- the authors hint at 4-6 participants being common for this type of analysis earlier in the text, which I think would be best placed in the sampling session including a little more detail for the rationale of choosing the number of 5 investigated here.

Table 1 patient demographics:

- The information in the table refers to 29 sessions being offered in 3 cases, the text says “up to 28”, which was confusing to me. It would be helpful to be consistent with the numbers or add an explanation why they differ.

- It would be interesting to place the characteristics of the 5 patients into the broader characteristics of the adolescents receiving STPP – e.g. were they similar in their baseline MFQ scores and age?

Data analysis

- I would find it helpful to elaborate on the data analysis also in the text, at least naming the steps of the iterative process, otherwise it is my opinion that the text is not comprehensive enough without consultation of the table (e.g. “Themes in this study were selected”)

- L.234 illustrates �extracts illustrate

Results

- I find the diverging results of Shelah and Mitch and the idea of the importance of adjusting to the psychoanalytic model very interesting and I can relate to the argument of it being a core element of the therapeutic process; however I cannot quite follow the argument as it is described in the text – could you elaborate here (with regard to the fact that they both still had good quantitative outcomes and at least Mitch continued to go)?

- Instead of “makes clear” (in lines 464, 504 and 506) I would recommend using a more cautious language when generalizing the individual adolescent’s experience (e.g. using “suggests”)

- In line 504, I was confused about the interpretation of trust being lost in Anayas therapy. For me, that was not implicated in the presented statements, could you elaborate?

- Line 507: I think the manuscript would benefit from an earlier introduction of a delineation of “successful therapeutic process” (which seems to be used for only three of the five therapies although they all had “good outcomes” – an issue that is not mentioned before the discussion section).

Discussion

- In addition to preventing ruptures, it seems to be the case in at least one of the therapies (Selah), that a rupture happened and was not repaired; In combination with her quite aversive statements (although her quantitative outcome was good), I would be interested in the discussion of her specific therapeutic process as successful or not, ideally additionally considering the therapist’s evaluation of her symptoms and therapeutic process (if this information Is available). Moreover, with regard to the growing literature on rupture and repair in therapy, I would add the need for repairing ruptures (instead of merely preventing them) as one important clinical implication of the study.

Reviewer #2: This article is a qualitative study based on data collected in the context of a RCT. Due to limits in confidentiality the data analyzed here (interview transcripts) could not be made available.

The article is a welcome contribution to the wider effort to identify "active ingredients" of psychotherapy and mechanisms of change. Presenting the perspective of service users and utilizing their own reports about their experience adds to our understanding of what they feel works in psychotherapy. Similar work has been done with adolescents of other therapeutic interventions (randomized in other legs of the IMPACT-ME study) and this article fills a gap by carrying out this work for adolescents randomized to receive STPP.

In terms of language, the article is well-organized and well-written, requiring only minor corrections:

Line 31: Sentence fragment

Line 85: Capitalize each of the first letter of cognitive behavior therapy for consistency

Line 170: “Equipping adolescent…are thought to contribute” should change into “Equipping adolescents…is thought to contribute”

Lines 465-468: awkward sentence structure

Line 504: Was “Anaya” meant to be “Selah”?

Regarding the implications of this study might the authors consider including as an implication that the results could be used to amend/enrich the STPP protocol, especially with regards to the way the treatment is framed so that the adolescents can better understand what to expect? For instance, anxiety in STPP seems to be both potentially facilitating or hindering for service users. If service users were better informed about what to expect in STPP as part of the STPP protocol, could that mitigate the potentially hindering effect of anxiety and ease the adolescent's process of adjustment to the particular treatment model?

Also, in including participants who showed good outcomes but who seemed to have a negative experience of the treatment, would the authors be willing to consider the possibility that their response might constitute support for (or be explained by) the idea of "flight into health," a form of resistance to psychotherapy?

Overall, the article makes a valuable contribution, not only in the filed of psychotherapy research but also in the area of clinical practice.

6. PLOS authors have the option to publish the peer review history of their article (what does this mean?). If published, this will include your full peer review and any attached files.

Reviewer #1: No

Reviewer #2: **Yes: **Yianna Ioannou, PhD

---

## [Author Response · Author response to Decision Letter 0]

25 Aug 2021

Please see separate document, uploaded with response to all reviewer comments.

---

## [Editor Report · Decision Letter 1]

31 Aug 2021

What contributes to good outcomes? The perspective of young people on short-term psychoanalytic psychotherapy for depressed adolescents

PONE-D-21-07634R1

Dear Dr. Midgley,

We’re pleased to inform you that your manuscript has been judged scientifically suitable for publication and will be formally accepted for publication once it meets all outstanding technical requirements.

Kind regards,

Svenja Taubner

Academic Editor

PLOS ONE
---

## [Editor Report · Acceptance letter]

13 Sep 2021

PONE-D-21-07634R1 

What contributes to good outcomes? The perspective of young people on short-term psychoanalytic psychotherapy for depressed adolescents 

Dear Dr. Midgley:

I'm pleased to inform you that your manuscript has been deemed suitable for publication in PLOS ONE. Congratulations! Your manuscript is now with our production department. 

Kind regards, 

on behalf of

Dr. Svenja Taubner 

Academic Editor

PLOS ONE